# Genotype Uniformity, Wild Bird-to-Poultry Transmissions, and Farm-to-Farm Carryover during the Spread of the Highly Pathogenic Avian Influenza H5N8 in the Czech Republic in 2021

**DOI:** 10.3390/v14071411

**Published:** 2022-06-28

**Authors:** Alexander Nagy, Lenka Černíková, Martina Stará, Lada Hofmannová, Kamil Sedlák

**Affiliations:** State Veterinary Institute Prague, Sídlištní 136/24, 165 03 Prague, Czech Republic; alexander.nagy@svupraha.cz (A.N.); lenka.cernikova@svupraha.cz (L.Č.); lada.hofmannova@svupraha.cz (L.H.); kamil.sedlak@svupraha.cz (K.S.)

**Keywords:** H5N8, HPAI, avian influenza, highly pathogenic avian influenza, outbreak, poultry

## Abstract

In 2020–2021, the second massive dissemination of a highly pathogenic avian influenza of the H5Nx subtype occurred in Europe. During this period, the virus caused numerous outbreaks in poultry, including in the Czech Republic. In the present study, we provide an insight into the genetic variability of the Czech/2021 (CZE/2021) H5N8 viruses to determine the relationships between strains from wild and domestic poultry and to infer transmission routes between the affected flocks of commercial poultry. For this purpose, whole genome sequencing and phylogenetic analysis of 70 H5N8 genomes representing 79.7% of the cases were performed. All CZE/2021 H5N8 viruses belonged to the 2.3.4.4b H5 lineage and circulated without reassortment, retaining the A/chicken/Iraq/1/2020 H5N8-like genotype constellation. Phylogenetic analysis suggested the frequent local transmission of H5N8 from wild birds to backyard poultry and extensive spread among commercial poultry farms. In addition, the analysis suggested one cross-border transmission event. Indirect transmission via contaminated materials was considered the most likely source of infection. Improved biosecurity and increased collaboration between field veterinarians and the laboratory are essential to limit the local spread of the virus and to reveal and interrupt critical routes of infection.

## 1. Introduction

Between October 2020 and June 2021, the highly pathogenic avian influenza (HPAI) H5N8 re-emerged in Europe [1]. In terms of severity, the 2020/2021 outbreak is comparable to the first severe H5N8 event in 2016/2017, including the number of countries affected and the number of domestic and wild bird species infected [1,2].

Available data suggest that the origin of the European H5N8 strains in 2020–2021 can be traced to the H5 lineage 2.3.4.4b virus detected during the HPAI infection in chickens in the Republic of Iraq in May 2020 [3]. Subsequently, the A/chicken/Iraq/1/2020-like H5N8 strains were detected in late July 2020 in Central and North Kazakhstan and the neighboring southern regions of Central Russia. The origin of the A/chicken/Iraq/1/2020 H5N8 strain itself is less obvious. However, sequence data suggest that the virus circulated undetected for a relatively long time period without reassortment and retained the genotype constellation similar to the Eurasian H5N8 strains from 2017/2018 [3]. In Europe, the A/chicken/Iraq/1/2020-like H5N8 virus was first detected in October 2020 in wild birds from the Netherlands, Germany, and the United Kingdom [3]. These detections were followed by a sharp increase in the reporting of H5N8 outbreaks starting in November 2020 [2], with rapid widespread dissemination affecting 27 countries. The pan-European spread was followed by reassortment with various low-pathogenic avian influenza viruses of Eurasian origin, resulting in five subtypes (H5N1, H5N3, H5N4, H5N5, and H5N8) and 19 distinct genotype constellations [2].

In the Czech Republic, the first H5N8 strain was discovered at the end of January 2021, followed by a sharp increase in outbreak reports between January and May 2021 in backyard and commercial poultry. This was accompanied by numerous detections of the H5N8 virus in wild birds, leading to the second massive spread of the disease since 2017 [4]. The aim of the study presented was to elucidate the transmission and spread of H5N8 in the Czech Republic. To this end, whole genome sequencing and phylogenetic analysis of 70 representative H5N8 viruses detected in backyard and commercial poultry, wild birds, and hobby/exotic birds between January and May 2021 were performed.

## 2. Materials and Methods

All bird carcasses were subjected to autopsy investigation. Cloacal and tracheal swabs or parts of multiple organs were collected for detection and next-generation sequencing. The pooled organs were homogenized in RNA later (Invitrogen, Waltham, MA, USA) and swabs in PBS, respectively. Total nucleic acid was extracted from 200 µL of supernatant of pooled organs and swabs (MagNAPure Compact, MagNAPure 24, or MagNAPure 96 instruments, Roche, Basel, Switzerland) and eluted to 50 µL. For influenza detection and subtype identification, RT-qPCR methods specific to generic influenza A virus and subtypes H5 and N8, in combination with cleavage site sequencing [5,6,7] were performed.

Specimens with Cq ≤ 30 were subjected to whole genome sequencing. Real-time next-generation sequencing was performed using the nanopore technology (MinION Mk1B, R9.1.4 flow cells; Oxford Nanopore Technologies, Oxford, UK). The H5N8 genome was amplified (OneStep RT-PCR Kit, Qiagen, Hilden, Germany) in a final reaction volume of 12.5 µL (10 µL RT-PCR mix + 2.5 µL total NA extract; primers available on request). Sequencing libraries were purified (SPRIselect beads; Beckman Coulter) and quantified (QIAxpert; Qiagen). End-preparation, native barcoding, and sequencing adapter ligation were performed according to the manufacturer’s instructions. Basecalling was performed with Guppy v4.40 and demultiplexing and reference mapping by implementing the RAMPART (Read Assignment, Mapping, and Phylogenetic Analysis in Real Time) module of the ARTIC bioinformatic pipeline [8] set to the concatenated H5N8 genome as a reference. Consensus sequences were obtained by samtools mpileup and bcftools utilities [9] and submitted to the GISAID EpiFlu database (Acc. No. EPI_ISL_956368, EPI_ISL_977513, EPI_ISL_980839, EPI_ISL_1033028-30, EPI_ISL_1033124, EPI_ISL_1057106, EPI_ISL_1058019-22, EPI_ISL_1080480, EPI_ISL_1080579, EPI_ISL_1180234, EPI_ISL_1191587, EPI_ISL_1399231-45, EPI_ISL_1697181-200, EPI_ISL_1941351, EPI_ISL_1941365, EPI_ISL_1941375, EPI_ISL_1941417, EPI_ISL_1941444, EPI_ISL_1941480, EPI_ISL_1941542, EPI_ISL_1941578-82, EPI_ISL_2194543, EPI_ISL_2402936-38, EPI_ISL_12436968, EPI_ISL_12437699, and EPI_ISL_12437700).

H5N8 genomes from the Czech/2021 outbreaks were compared with other European sequences collected between November 2020 and 2021 and stored in the GISAID database. The sequences were aligned using the MAFFT web server (multiple alignment using fast Fourier transform [10]). Sequence difference count matrices at the nucleic and amino acid levels were calculated using the BioEdit 7.0.9.0 program [11]. Sequence trimming and format conversion (Phylip full names and padded) were performed using the AliView program [12]. Maximum likelihood (ML) trees and phylogenetic dating were performed using the IQ-TREE multicore version 2.2.0 for Linux 64-bit [13]. ML trees (1000 replicates) were calculated separately for each genomic segment (Appendix A) and as a species correlation tree constructed from genomic concatenates. Sequence concatenation was performed using union from the EMBOSS applications [14]. For all trees, the best fitting models were selected according to the Bayesian information criterion (BIC). The concatenated species correlated ML tree rooted to the A/chicken/Iraq/1/2020 H5N8 strain was used for phylogenetic dating via IQ-TREE by implementing the least square dating (LSD2) method [15]. All trees were visualized in FigTree v1.4.4. Sequence features and antiviral resistance markers were investigated by using the bioinformatic resources of the Influenza Research Database [16].

## 3. Results

In the Czech Republic, the H5N8 season started on 21 January and lasted until 18 May 2021 (Calendar Weeks 3–20; Figure 1, Table 1). The dominant HPAI subtype was H5N8, which was identified in the overwhelming majority (98.5%) of localities. One HPAI of the H5N5 subtype was detected in a mute swan carcass. A total of 37 H5N8 outbreaks were identified in poultry, including 24 backyard farms and 13 commercial farms (12 duck and one chicken). In addition, H5N8 was detected in peacocks (*Pavo cristatus*), and the disease was also transmitted to an Australian brushturkey (*Alectura lathami*). In the meantime, the H5N8 subtype was detected in 51 wild birds from 20 localities: 40 mute swans (*Cygnus olor*), seven mallards (*Anas platyrhynchos*), one graylag goose (*Anser anser*), one common coot (*Fulica atra*), and one white stork (*Ciconia ciconia*). A total of 271,328 specimens of poultry were destroyed during management of the outbreak. An overview of the pathological macroscopic lesions associated with an HPAI infection is provided in Appendix A.

Figure 1 shows the temporal distribution of the number of outbreaks identified in wild birds, backyard and commercial poultry, and hobby/exotic birds per calendar weeks in the Czech Republic. As can be seen, the outbreaks in wild birds and backyard poultry occurred throughout the entire period, while the H5N8 in commercial poultry was detected mainly in Calendar Weeks 11–12 (10 out of 13 farms), followed by three additional outbreaks during Weeks 14–15. Mapping of the outbreaks (Figure 1) revealed a relatively widespread distribution of the H5N8 virus in both wild birds and backyard poultry, with frequently adjacent or overlapping localities. Of particular interest is a relatively dense outbreak cluster formed in the north-central part of the country. Here, 11 out of 12 commercial Pekin duck breeding farms belonging to the same company and a commercial hen farm for egg production of a different owner were affected by the H5N8 infection. All these farms are located within a circle with approximately 30 km in diameter. The twelfth Pekin duck breeding farm was situated in an easterly direction at a distance of approximately 50 km.

To investigate the relationships between the H5N8/2021 strains circulated during the HPAI epidemic in the Czech Republic, 70 viral genomes were sequenced and analyzed. H5N8 genome sequences were obtained from 65% (13/20) of wild bird localities, 87.5% (21/24) of domestic flocks, and 84.6% (11/13) of commercial farms. In addition, the data include H5N8 genomic information from peacocks and an Australian brushturkey. Overall, H5N8 genomic sequence information is available from 79.7% (47/59) of all cases. All the CZE/2021 H5N8 hemagglutinin (HA) sequences had the amino acid motif REKRRKR/GLF in the cleavage sites, indicating a highly pathogenic phenotype in chickens. No additional mutations conferring antiviral resistance, altering receptor binding preference, or increasing virulence or replication ability in mammals were observed. Phylogenetic analysis of the particular genomic segments in the context of available Eurasian H5N8 genomes revealed that all segments of the CZE/2021 H5N8 strains belonged to the Iraq/1/2020-like subclade of lineage 2.3.4.4b (Appendix A). This suggests that CZE/2021 H5N8 viruses retained the “Iraqi” genotype constellation and circulated without reassortment.

To further investigate the relationships between the Czech H5N8 outbreaks, a maximum likelihood (ML) species correlation tree and least-squares dating were calculated from concatenated CZE/21 H5N8 genomes. Phylogenetic analysis revealed the evolution of the CZE/2021 genomes along three clearly discernible sublineages (Figure 2). In all sublineages, genomes from wild birds are frequently grouped with those from backyard poultry and hobby/exotic birds. This suggests the frequent local transmission of the H5N8 from the wild bird reservoir.

H5N8 in commercial holdings was initially detected between 18 and 19 March 2021 in five Pekin duck breeding farms established for egg production containing 3000–26,000 birds (Table 1). A common sign of infection was the dramatic ~80% decrease in egg production, while the mortality remained relatively low, below 1%. The ducks were apparently vital, as no decrease in food or water intake was observed. Two birds in a single farm showed clinical signs with a restricted ability to move. Post-mortem examination of the birds concerned showed a generally good condition with an enlarged spleen and serous pericarditis as the only necropsy findings that could have been associated with HPAI infection. All of the affected farms were quickly depopulated, and 3 km protective zones and 10 km surveillance zones were established. Subsequent investigation of cloacal and tracheal swabs collected from neighboring breeding flocks between 26 and 29 March 2021 and between 7 and 18 April 2021 revealed, respectively, four and three H5N8-positive Pekin duck farms containing 1800–7800 birds. According to the available data, these farms did not show remarkable signs of infection and were only discovered during the monitoring activities carried out in the defined zones. Finally, an increased mortality in hens reared for chicken egg production was observed on 29 March 2021, with subsequent confirmation of H5N8 positivity. The hen farm belonged to another company located near the infected duck farms.

According to the zootechnical records, the breeding of Pekin ducks in the company concerned proceeds according to the following schedule. Young, day-old ducklings (Cherry Valley Pekin ducks) are imported from Germany (Karlsdorf) every three weeks, separated into males and females, and bred for three weeks in breeding houses in a single location. Thereafter, males and females are moved and fattened separately in multiple fattening farms for an additional 16 weeks. Finally, the 19-week-old birds are mixed in a 1:5 male-to-female ratio and moved to the final stables designated for egg production, where the birds usually spend 50 weeks. All H5N8 infections in commercial ducks were identified in fattening farms or terminal stables for egg production.

H5N8 genomes were obtained from all five farms detected at the beginning of the outbreak, i.e., on 18 and 19 March 2021, two H5N8 genomes were sequenced from five farms detected between 26 and 29 March 2021, and representative H5N8 genomes were obtained from all three remaining farms detected as positive between 7 and 18 April 2021. In total, sequence information was obtained from 11 out of 13 (84.6%) affected farms (Table 1). The concatenated genomic ML tree revealed that all H5N8 viruses collected from commercial ducks and chickens showed high nucleotide identity (>99.7%) and belonged to one well-defined subclade (Figure 2). Furthermore, the ML tree showed that two distinct phylogenetic groups, 3A and 3B, were already present at the time of outbreak identification. LSD analysis placed the most recent common ancestor (MRCA) of the entire subclade in late December 2020, while the MRCAs of Groups 3A and 3B were placed two months later, during late February and early March 2021, respectively (Figure 3). In general, the time lapse between the positive farm identification and the particular MRCA ranged from one to three weeks. In Group 3A, the trunk position of the index strain CZE/5466/21 suggests that this farm can be considered a common ancestor for three second-order groups 3A1–A3, with the MRCA circulated around the end of March 2021. In addition, within Group 3A2, there was an apparent third order chicken-to-duck farm transmission, with the MRCA placed around the middle of March 2021 (Figure 3). These findings allowed us to deduce the most putative farm-to-farm transmission routes shown in Appendix A.

The relationships between the farms included in Group 3B are less clear. Group 3B comprises six Pekin duck farms with one well-supported second-order group 3B1, including a single transmission event to backyard chickens (Table 1, ID 6527/21). The MRCA of Subgroup 3B1 is uncertain. Interestingly, Group 3B also encompasses the A/chicken/Romania/12448_21VIR3734-3/2021 H5N8 strain detected in commercial chickens on 5 May 2021 in Ungheni, Mures County, located in the middle of Romania. The relationship of the Romanian strain to the Czech commercial duck H5N8 genomes was first observed in the phylogenetic trees calculated separately for individual genomic segments in the context of the available Eurasian H5N8 genomes (Appendix A). Similarly, in the concatenated ML tree, the Romanian H5N8 strain was clearly assigned to Group 3B with high nucleotide sequence identity (Figure 2 and Figure 3) and the MRCA placed at the beginning of March 2021.

## 4. Discussion

Phylogenetic analysis of CZE/H5N8 genomes suggested that the emergence of infection followed the same modus operandi observed in previous H5Nx HPAI seasons, i.e., multiple incursions of co-circulating variants via wild bird vectors and secondary spread and transmission to backyard and commercial poultry [2,4,17,18,19]. In general, transmission of HPAI to poultry is influenced by multiple risk factors acting at the local level [20], the most critical of which are the density of wild bird populations, the proximity of wetlands [21], the distance between poultry houses [22,23,24], species composition, and the type of rearing. Another important aspect of disease transmission is negligence or the loose implementation of biosecurity and preventive measures combined with low levels of surveillance. Many of these conditions have also been observed in the particular outbreaks in the Czech Republic. Given the local setting of critical factors and analyses of previous H5N1 and H5N8 outbreaks, high-risk areas and geographical foci for the spread of HPAI have been inferred [25,26]. Interestingly, the index strain of the Czech 2021 H5N8 outbreak (21 January 2021, mute swan carcass ID 1410/21) and the first outbreak in backyard poultry (22 January 2021, chicken ID 1566/21) were also identified in one of these areas in the Czech Republic [4]. Available data suggests that avian influenza activity in these areas may be high, with intensive reassortment events [27]. Targeted surveillance for avian influenza in wild bird populations in those hotspots would therefore reveal the HPAI entering the region well before the first outbreaks appear in the backyard farms. Early warning to local farmers would save the total cost incurred for the full spectrum of descendent activities necessary to eradicate the disease, such as depopulation, disinfection, the establishment and monitoring of protection and surveillance zones, and repopulation [28].

During 2021, H5N8 was also transmitted to farmed Pekin ducks, representing one of the most serious infections of commercial poultry ever recorded in the Czech Republic. It is difficult to infer the origin of the infection and to reconstruct the sequence of transmission events between the affected farms in detail. This is because of virus detection in multiple localities within a short time frame as well as mild or even subclinical infections. Moreover, the relationships between the particular farms in terms of the transport of materials, birds, eggs, feed, and other accessories are unknown. Other problems included low sample positivity, a lack of relevant field information, and the absence of a comprehensive sampling strategy in affected flocks. 

The sudden discovery of the two phylogenetically distinct H5N8 groups 3A and 3B suggests that the virus had been circulating undetected long prior to the outbreak identification and had been transmitted to the first positive farms in at least two independent transmission events. Accordingly, the MRCA of both groups was placed in late December 2020. The grouping of all H5N8 strains from commercial ducks into the same subclade implies that the infection, rather than being transmitted multiple times from a wild bird reservoir, originated from a common source, likely a Pekin duck farm. This index farm has not been identified. However, the occurrence of outbreaks exclusively in fattening or final stables places the origin in the Czech Republic and excludes transmission from Germany. Even the ML trees of the available European H5N8 strains, including 154 virus sequences from Germany, do not suggest importation of the disease. Therefore, we assumed routes of transmission other than those related to stable restocking. Similarly, the division of Groups 3A and 3B into second- or even third-order groups in a short time frame strongly suggests an indirect spread of the disease, probably via contaminated materials, and implicates a whole network of contacts. This is evident from commercial duck-to-duck, commercial chicken-to-duck, and commercial duck-to-backyard chicken transmissions observed in the phylogenetic tree. Although sequence information from key positive farms may be lacking within Group 3A, we were able to reconstruct the most likely and chronologically sound transmission routes, indicated in the Appendix A.

Further, the common origin of Group 3B CZE/2021 H5N8 viruses and the A/chicken/Romania/12448_21VIR3734-3/2021 H5N8 strain suggests cross-border transmission. The Romanian H5N8 strain was detected on 5 May 2021, i.e., approximately two weeks after the last commercial outbreak in the Czech Republic. According to Czech veterinary records, the cross-border transmission suspected from the phylogenetic analysis can be associated with three export events of one-day-old ducklings. All exports from the Czech Republic to Romania directed to the municipality of Giarmata in Temes County on 4, 11, and 17 March 2021, i.e., shortly before the identification of the outbreak in Romanian commercial chickens. The birds were transported in cardboard boxes, and the lorries were carefully disinfected in Romania after picking up the birds and after their return to the Czech Republic. Moreover, the lorry drivers were different from those who regularly operate between commercial farms. Further, the records also showed that the eggs from which the ducklings hatched came from several farms, and all three exports included birds hatched from eggs taken from at least one farm included in Group 3B. Similarly, the date of transmission correlates strongly with the MRCA of Group 3B, placed at early March 2021. However, Giarmata is located ~250 km in a direct line from Ungheni. Moreover, the only H5N8 outbreak in 2021 in Temes County occurred in backyard poultry on 24 February 2021, i.e., before the first export from the Czech Republic took place [29]. Therefore, there is no direct link between the Group 3B CZE/H5N8/2021 strains and the outbreak in commercial chickens in Ungheni in Romania. This definitely rules out the direct transmission of the H5N8 virus via exports from the Czech Republic. The origin of phylogenetic relationships between the Group 3B CZE/H5N8/2021 and the A/chicken/Romania/12448_21VIR3734-3/2021 H5N8 strains remains unexplained.

The phylogenetic and spatio-temporal analysis revealed key aspects that promoted the efficient spread of HPAI infection among commercial poultry farms, the most critical of which were the species of birds and the density and mutual proximity of the affected farms. Several studies suggest the pivotal role of domestic ducks in the maintenance and spread of HPAI viruses [17,29,30]. The proximity of duck farms is often associated with an increased risk of HPAI introduction into gallinaceous poultry. In domestic ducks, the H5N8 virus can cause infections with mild clinical signs [30,31,32] and can thus circulate undetected for a relatively long time. Similarly, in Czech-farmed ducks, H5N8 was initially detected in flocks set up for egg production, with the only recorded symptom associated with infection being a decline in egg laying. The mortality rate was considered insignificant, and the only observed clinical signs could not at first sight be associated with HPAI infection. In addition, most of the duck farms were identified as being H5N8-positive only during monitoring activities, because they were spanned within surveillance or protective zones. A dramatic decrease in egg production and a low mortality of breeding ducks were also observed during the H5N8 event in 2019–2020 in Poland [18]. However, in these cases, the infection was accompanied by a decrease in food and water intake, which was not observed in the Czech Pekin duck holdings.

In addition to the mild infection of ducks, another critical factor is the local farm density [24], which seems to have contributed significantly to the spread of HPAI among the Czech commercial farms. Twelve of the 13 affected farms were situated within a circle of approximately 30 km in diameter. All Pekin duck farms belong to the same company. In addition, all farms were relatively densely populated and contained 1800–26,200 birds. The chicken farms, which belonged to a different company and were located within the same radius, contained 173,400 chickens.

In summary, the outbreaks in commercial poultry in the Czech Republic were caused by a constellation of two main adverse factors: a subclinical infection of Pekin ducks and the high local density of poultry farms with intensive mutual contacts. The human-mediated indirect spread of the HPAI between commercial poultry farms is a frequently observed phenomenon during HPAI outbreaks [18,22,23,33]. However, in the case of Czech commercial poultry outbreaks, human-assisted spread might have been exacerbated by the culmination of the SARS-CoV-2 epidemic, which led to a reduction in human resources required for bird keeping and other flock management activities and a loose implementation of biosecurity measures.

Real-time next-generation sequencing and phylogenetic analysis of influenza virus genomes can significantly complement epidemiological studies during HPAI outbreaks and allow the identification of infection clusters and specific farm-to-farm transmission events. However, to effectively manage HPAI infections in the future, specifically to monitor and prevent the local spread of the virus and to detect and interrupt critical infectious pathways, increased collaboration between field and laboratory staff in terms of data sharing and operative sampling is required.

## Figures and Tables

**Figure 1 viruses-14-01411-f001:**
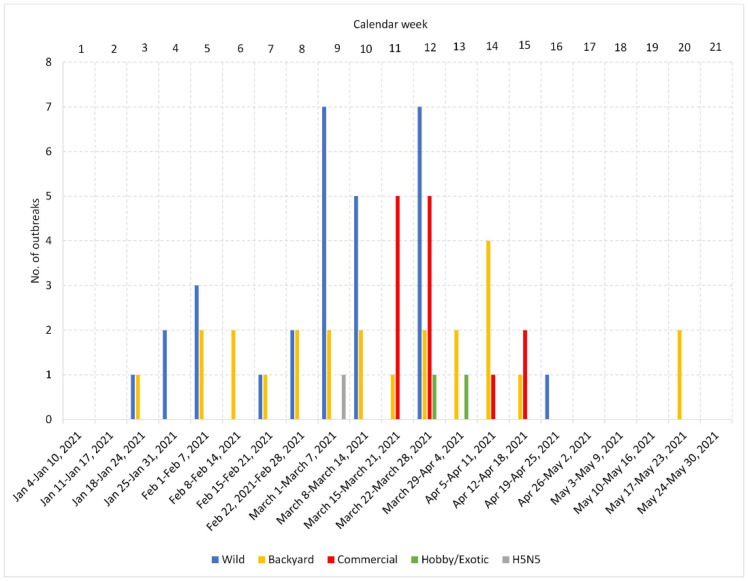
Distribution of HPAI outbreaks in calendar weeks and geographical locations. Bar chart showing the cumulative number of HPAI outbreaks in each calendar week in the Czech Republic. Geographical distribution of the outbreak localities is provided in the map available online at https://www.google.com/maps/d/u/0/viewer?mid=1MNHHITwIvLUs0sk36Z75OjPGt9h98OqZ&fbclid=IwAR2nXOIkdSAZDUeddMQXcYQH5Y33Yyb2cd5cQ46Ew1dGmjVCFSk_wPhV1e0&ll=50.22443150000001%2C15.479471699999987&z=8 (accessed on 28 May 2022).

**Figure 2 viruses-14-01411-f002:**
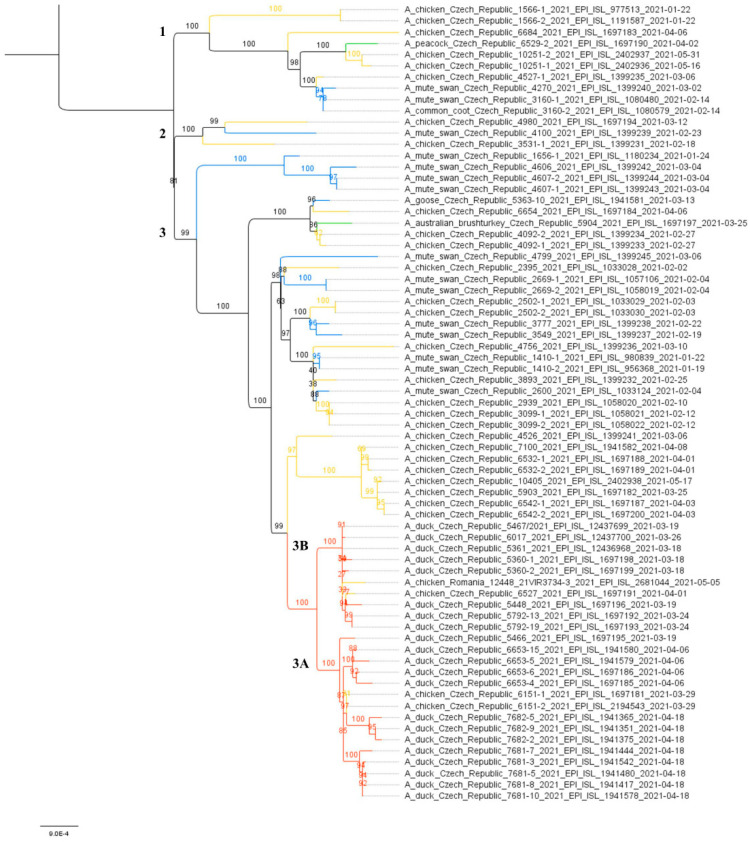
Species correlation tree. The ML tree was calculated on the basis of 70 concatenated CZE/2021 H5N8 genomes using the IQ-TREE (K3Pu + F+G4 as the best fit model selected according to the Bayesian information criterion). The tree was drawn to scale with branch lengths measured in the number of substitutions per site, and the bootstrap values (1000 replicates) were shown in percentages. Tree clades were colored according to origin: red—commercial; orange—backyard; blue—wild; green—hobby/exotic. The subclades are designated with the numbers 1–3. A subtree including H5N8 strains from commercial poultry is shown in more detail in Figure 3.

**Figure 3 viruses-14-01411-f003:**
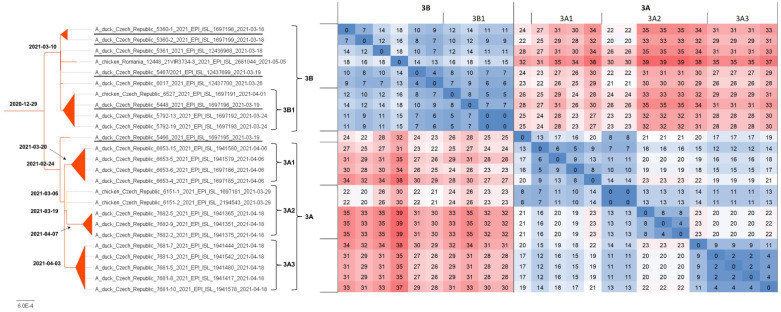
Phylogenetic dating. Concatenated ML tree with the most probable divergence time estimated using the LSD approach (bold). The tree was rooted to the A/chicken/Iraq/1/2020 H5N8 strain. For clarity, only the subtree including H5N8 strains from commercial poultry (see Figure 2) was presented along with a nucleotide sequence difference count matrix in absolute values as a heatmap.

**Table 1 viruses-14-01411-t001:** The 2021 H5N8 outbreak localities in the Czech Republic listed in chronological order.

Sample ID ^#^	Collection Date	Subtype	Category	District	Locality	Bird Species/No.	Genomic Sequences *
1410/21	1/21/2021	H5N8	Wild	Písek	Zlivice	Mute swan 2×	2
1566/21	1/22/2021	H5N8	Backyard poultry	Tábor	Dlouhá Lhota u Tábora	Chicken 32×	2
1655/21	1/23/2021	H5N8	Wild	Písek	Hradiště u Písku	Mute swan 1×	n.a.
1656/21	1/25/2021	H5N8	Wild	České Budějovice	Haklovy Dvory	Mute swan 1×	1
						Wild duck 5×	n.a.
2395/21	2/2/2021	H5N8	Backyard poultry	Benešov	Jankov	Chicken 53×	1
2502/21	2/3/2021	H5N8	Backyard poultry	Chrudim	Ronov n. Doubravou	Chicken 61×	2
2600/21	2/4/2021	H5N8	Wild	Strakonice	Strakonice	Mute swan 1×	1
2669/21	2/5/2021	H5N8	Wild	Tábor	Soběslav	Mute swan 2×	2
2939/21	2/10/2021	H5N8	Backyard poultry	Strakonice	Katovice	Chicken 57×	1
3099/21	2/12/2021	H5N8	Backyard poultry	Plzeň—jih	Bzí	Chicken 21×	2
3160/21	2/16/2021	H5N8	Wild	Liberec	Příšovice	Mute swan 5×	1
						Common coot 2×	1
3531/21	2/19/2021	H5N8	Backyard poultry	Pelhřimov	Horní Cerekev	Chicken Geese 443× ^#^	1
3549/21	2/22/2021	H5N8	Wild	Nymburk	Nymburk	Mute swan 1×	1
3893/21	2/25/2021	H5N8	Backyard poultry	Strakonice	Rojice	Chicken 17×	1
3777/21	2/26/2021	H5N8	Wild	Praha—východ	Květnice	Mute swan 1×	1
4092/21	2/27/2021	H5N8	Backyard poultry	Plzeň—sever	Hněvnice	Chicken 51×	2
4098/21	3/3/2021	H5N8	Wild	Zlín	Zlín	Wild duck 2×	n.a.
4099/21 ^&^	3/3/2021	H5N5	Wild	Kroměříž	Kvasice	Mute swan 1×	1
4100/21	3/3/2021	H5N8	Wild	Jeseník	Javoní-Ves	Mute swan 1×	1
4125/21	3/3/2021	H5N8	Wild	Praha—město	Smíchov	Mute swan 3×	n.a.
4193/21	3/4/2021	H5N8	Wild	Louny	Žatec	Mute swan 1×	n.a.
4270/21	3/4/2021	H5N8	Wild	Mladá Boleslav	Kněžmost	Mute swan 1×	1
4271/21	3/4/2021	H5N8	Wild	Mladá Boleslav	Kněžmost	Mute swan 5×	n.a.
4526/21	3/6/2021	H5N8	Backyard poultry	Opava	Velké Hoštice	Chicken 103×	1
4527/21	3/6/2021	H5N8	Backyard poultry	Mladá Boleslav	Březno u Mladé Boleslavi	Chicken 22×	1
4606/21	3/10/2021	H5N8	Wild	Uherské Hradiště	Chylice	Mute swan 1×	1
4607/21	3/10/2021	H5N8	Wild	Olomouc	Chomoutov	Mute swan 2×	2
4608/21	3/10/2021	H5N8	Wild	Olomouc	Hynkov	Mute swan 1×	n.a.
4756/21	3/10/2021	H5N8	Backyard poultry	Příbram	Březnice	Chicken 35×	1
4799/21	3/12/2021	H5N8	Wild	Opava	Kateřinky u Opavy	Mute swan 1×	1
4980/21	3/13/2021	H5N8	Backyard poultry	Uherské Hradiště	Osvětimany	Chicken 406×	1
5363/21	3/13/2021	H5N8	Wild	Plzeň—mesto	Plzeň	Greylag goose 1× Mute swan 10×	1 n.a.
5146/21	3/17/2021	H5N8	Backyard poultry	Strakonice	Blatná	Chicken 40×	n.a.
5360/21	3/18/2021	H5N8	Commercial breeding ducks	Hradec Králové	Dobřenice	Pekin duck 26,264×	2
5361/21	3/18/2021	H5N8	Commercial breeding ducks	Hradec Králové	Chlumec n. Cidlinou	Pekin duck 14,479×	1
5448/21	3/19/2021	H5N8	Commercial breeding ducks	Nymburk	Vinice u Městce Králové	Pekin duck 2970×	1
5466/21	3/19/2021	H5N8	Commercial breeding ducks	Pardubice	Vápno u Přelouče	Pekin duck 3346	1
5467/21	3/19/2021	H5N8	Commercial breeding ducks	Hradec Králové	Chudonice	Pekin duck 6550	1
5867/21	3/25/2021	H5N8	Backyard poultry	Praha—východ	Čelákovice	Chicken 40×	n.a.
5903/21	3/25/2021	H5N8	Backyard poultry	Jičín	Vysoké Veselí	Chicken 3×	1
5904/21	3/25/2021	H5N8	Hobby/exotic	Plzeň-město	Plzeň	Australian bushturkey	1
5791/21	3/26/2021	H5N8	Commercial breeding ducks	Hradec Králové	Kosičky	Pekin duck 1832×	n.a.
5792/21	3/26/2021	H5N8	Commercial breeding ducks	Hradec Králové	Luková n. Cidlinou	Pekin duck 3347	2
6017/21	3/26/2021	H5N8	Commercial breeding ducks	Nymburk	Slibovice	Pekin duck 7086	1
6039/21	3/29/2021	H5N8	Commercial breeding ducks	Nymburk	Záhornice u Městce Králové	Pekin duck 7537	n.a.
6151/21	3/29/2021	H5N8	Commercial chickens	Hradec Králové	Kosičky	Chicken 173,394	2
6527/21	4/1/2021	H5N8	Backyard poultry	Hradec Králové	Dobřenice	Chicken 34×	1
6532/21	4/1/2021	H5N8	Backyard poultry	Bruntál	Loučky u Zátoru	Chicken 15×	2
6529/21	4/3/2021	H5N8	Hobby/exotic	Teplice	Razice	Peacock	1
						Ostrich, all 43×	n.a.
6542/21	4/6/2021	H5N8	Backyard poultry	Praha—západ	Třebotov	Chicken 14×	2
6654/21	4/6/2021	H5N8	Backyard poultry	Tachov	Vysočany u Boru	Chicken 16×	1
6653/21	4/7/2021	H5N8	Commercial ducks	Hradec Králové	Starý Bydžov	Pekin duck 9385×	4
6684/21	4/7/2021	H5N8	Backyard poultry	Most	Polerady	Chicken 5×	1
6987/21	4/9/2021	H5N8	Backyard poultry	Beroun	Hodyně u Skuhrova	Chicken Duck 81× ^#^	n.a.
7100/21	4/12/2021	H5N8	Backyard poultry	Strakonice	Vodňany	Chicken 9×	1
7681/21	4/18/2021	H5N8	Commercial breeding ducks	Hradec Králové	Klamoš	Pekin duck 5600	5
7682/21	4/18/2021	H5N8	Commercial breeding ducks	Hradec Králové	Nové Město n. Cidlinou	Pekin duck 7779	3
7769/21	4/20/2021	H5N8	Wild	Strakonice	Vodňany	White stork 1×	n.a.
10251/21	5/17/2021	H5N8	Backyard poultry	Mělník	Chorušice	Chicken	2
10405/21	5/18/2021	H5N8	Backyard poultry	Mladá Boleslav	Čejetice u Mladé Boleslavi	Chicken	1

^&^ Not included in the study. * n.a. = not available. ^#^ Cumulative number of all birds.

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
