# Peer review of "Genotype Uniformity, Wild Bird-to-Poultry Transmissions, and Farm-to-Farm Carryover during the Spread of the Highly Pathogenic Avian Influenza H5N8 in the Czech Republic in 2021"

_viruses, 2022, doi:10.3390/v14071411_

Round 1

Reviewer 1 Report

Nagy et al. describe the genetic variability of HPAI H5N8 viruses detected in the Czech Republic in 2021, including phylogenetic analysis of transmission routes via full-genome sequencing. The paper holds valuable information, however some areas must be clarified before publication.

Material and Methods:

Line 75. In the abstract, the authors state that 70 full-genome sequences were created, however the accession numbers do not match (there are only 15 accession numbers given)?

Results:

Table 1. Maybe instead of “Strain” the authors could use “Sample ID” as the header as this is more fitting.

Figure 1. It would be helpful if Figure 1a and 1b were larger to enable easier detection of the information in the figure.

Figure 2/3. If possible, it would be helpful to try and keep the nomenclature of the strains/subclades the same. It is very confusing to see 1, 2 and 3 in Figure 2 and then A and B with numbers in Figure 3. More and defined descriptions of what is what would greatly improve the readers’ understanding.

Line 176. As the ducklings are imported, were any comparisons to Germany/European sequence data to exclude this transmission route?

Figure S3. What was used as a root? The way the trees are shown makes it difficult to see the genetic relations. A tree without so many different viruses but then shown as the whole tree would be better.

Author Response

Dear editor,

We enclose our response to the reviewers' comments on manuscript ID: viruses-1775847. The vast majority of comments and suggestions have been incorporated into the manuscript.

The main changes in the manuscript suggested by the reviewers are:

  1. Redrawing of Fig 1
  2. Change in Table 1
  3. Removal of the supplementary videos
  4. Inclusion of complete trees of European H5N8 segments in the supplementary data

A detailed response to each comment or suggestion is provided below in italics.

We are grateful for all suggestions as they have greatly improved the quality of our manuscript.

Sincerely,

Alexander Nagy

Rev#1

Material and Methods:

Line 75. In the abstract, the authors state that 70 full-genome sequences were created, however the accession numbers do not match (there are only 15 accession numbers given)?

We apologize for this oversight. The acc nos were supplemented:

EPI_ISL_956368, EPI_ISL_977513, EPI_ISL_980839, EPI_ISL_1033028-30, EPI_ISL_1033124, EPI_ISL_1057106, EPI_ISL_1058019-22, EPI_ISL_1080480, EPI_ISL_1080579, EPI_ISL_1180234, EPI_ISL_1191587, EPI_ISL_1399231-45, EPI_ISL_1697181-200, EPI_ISL_1941351, EPI_ISL_1941365, EPI_ISL_1941375, EPI_ISL_1941417, EPI_ISL_1941444, EPI_ISL_1941480, EPI_ISL_1941542, EPI_ISL_1941578-82, EPI_ISL_2194543, EPI_ISL_2402936-38, EPI_ISL_12436968, EPI_ISL_12437699, EPI_ISL_12437700

Results:

Table 1. Maybe instead of “Strain” the authors could use “Sample ID” as the header as this is more fitting.

Corrected

Figure 1. It would be helpful if Figure 1a and 1b were larger to enable easier detection of the information in the figure.

Figure 1a has been enlarged and redrawn to better illustrate the calendar weeks. Figure 1b has been removed and a link to an interactive map has been provided.

Figure 2/3. If possible, it would be helpful to try and keep the nomenclature of the strains/subclades the same. It is very confusing to see 1, 2 and 3 in Figure 2 and then A and B with numbers in Figure 3. More and defined descriptions of what is what would greatly improve the readers’ understanding.

Figures 2 and 3 have been redrawn. In Fig 2 the sub-clades were clearly indicated. In addition, the groups A and B in Fig 3 have been renamed as 3A and 3B including sub-groups, to maintain consistency with Fig 2. The nomenclature has also been corrected in the text.

Line 176. As the ducklings are imported, were any comparisons to Germany/European sequence data to exclude this transmission route?

The sequences were compared to available European H5N8 genomes collected since Nov 2020 (n>500, for a detailed counts please refer to the caption in Fig S1). Lines 82-83, 221-224 and 273-275.

Figure S3. What was used as a root? The way the trees are shown makes it difficult to see the genetic relations. A tree without so many different viruses but then shown as the whole tree would be better.

All trees except N8 were rooted to A/goose/Guangdong/1/1996 H5N1. The N8 tree was rooted to A/mallard/Czech Republic/14516/2007 H3N8. This information has been added in the caption of Figure S. The supplementary data have been redrawn to include the whole segment trees.

Reviewer 2 Report

In this article, Nagy and colleagues propose an epidemiological study focusing on the spreading of H5N8 strains of IAV during the outbreak 2021 in Czech Republic. The paper is well written but some minor and major points have to be raised before publication.

Line 95-96: “domestic poultry” the authors often used term as domestic poultry, commercial flocks etc. it is correct but for clarity they should precise each time if they talk about chickens or ducks in the text. This point is very important to clearly understand the transmission dynamcis between wild birds > ducks > chickens.

Table S1: Mallard are ducks and ducks are? Pekin ducks obviously but I think that you should precise the full name of the birds.

Line 97-101: from where are coming the peacock and Australian brushturkey? It is referred as hobby/exotic in table 1. Is it a zoo?? (I have noticed that it is written in the legend of Figure 1, replace exotic by zoo) And there is also an ostrich but it is only appearing in the table 1 and nowhere else? 51 wild bird carcasses were collected but only 37 in the table. No clinical sign for ostrich, white stork or Australian brushturkey?

Table 1: replace wild bird species by “bird species” and precise when it is ducks or chickens for each sample collected for backyard poultry.

Where is the link for the interactive map? The map in figure 1 is nice but weakly informative because there is no chronological information associated with the different dots and once again the discrimination between ducks and chickens does not appear (for domestic = orange dots)… I hope it is the case in the interactive map.

What is the correspondence between map 1 and the map in the supplementary data? Not much is missing to make it easier for the reader to understand and follow the spreading of the virus. Maybe it is in the interactive map.

In general, the manuscript is well written, but some links are missing between the text and the figures, it has to be improved. Check also the figures in the manuscript (51 wild bird carcasses but 37 described in the table, 70 genomes but 68 in the table.. maybe I’m mistaken).

Concerning the phylogenetic trees, in figure 3 the authors discriminate between group A and B.. where are this group in figure 2 tree?

Line 146: backyard poultry here refers to chickens… is it always the case all along the manuscript?

“Another important aspect of disease transmission to poultry is negligence or loose 228

implementation of biosecurity and preventive measures combined with low levels of sur- 229

veillance”. Yes, it is exemplified in the videos provided in supplementary data when you see people among H5N8-infected ducks without any biosecurity equipment (video2) and a guy trying to kick a duck to make it move in video 3 without any biosecurity equipment…. I think you should not present these videos!

Finally, I regret that the authors only analyze in viruses sequence the presence of the polybasic site of HA. What about other known markers? zoonotic potential? 

For instance, in the 2016-2017 "wave" Polish and Czech H5N8 did express a full length PB1-F2 while other european H5N8 mainly expressed a truncated form (non functional, which is exceptional in the avian reservoir). What about these H5N8 viruses? 

I d'ont think that you can obtain any information from the phylogenetic trees presented in supplementary data. Change the format to make them readable.

Author Response

Dear editor,

We enclose our response to the reviewers' comments on manuscript ID: viruses-1775847. The vast majority of comments and suggestions have been incorporated into the manuscript.

The main changes in the manuscript suggested by the reviewers are:

  1. Redrawing of Fig 1
  2. Change in Table 1
  3. Removal of the supplementary videos
  4. Inclusion of complete trees of European H5N8 segments in the supplementary data

A detailed response to each comment or suggestion is provided below in italics.

We are grateful for all suggestions as they have greatly improved the quality of our manuscript.

Sincerely,

Alexander Nagy

Rev#2

Line 95-96: “domestic poultry” the authors often used term as domestic poultry, commercial flocks etc. it is correct but for clarity they should precise each time if they talk about chickens or ducks in the text. This point is very important to clearly understand the transmission dynamics between wild birds > ducks > chickens.

We attempted to unify the terminology throughout the manuscript and used the term “commercial poultry” to both commercial ducks and chickens. Where we refer specifically to ducks or chickens, the term “commercial duck” or “commercial chicken” was used.

The study also includes numerous reports of H5N8 from small domestic farms from villages across the Czech Republic. They are raised in backyards and contain mostly chickens. Only one such small farm included both chickens and ducks (ID 6987/21 in Table 1). The sequence information from this particular locality is not available. For all these localities we used the term “backyard poultry”. Please, see also the corrections to Table 1 made  as a result of the Rev#2 suggestion below.

Table S1: Mallard are ducks and ducks are? Pekin ducks obviously but I think that you should precise the full name of the birds.

The mallards were changed to “Wild duck” and duck to “Domestic duck”

Line 97-101: from where are coming the peacock and Australian brushturkey? It is referred as hobby/exotic in table 1. Is it a zoo?? (I have noticed that it is written in the legend of Figure 1, replace exotic by zoo) And there is also an ostrich but it is only appearing in the table 1 and nowhere else? 51 wild bird carcasses were collected but only 37 in the table. No clinical sign for ostrich, white stork or Australian brushturkey?

From the brushturkey we obtained only specimens from organs. This bird was not examined in our institute and the metadata were not available. The ostrich and a peacock came from a hobby farm. The peacock specimen was successfully sequenced. The ostrich was only weakly positive and the WGS was not successful. From the white stork we obtained only cloacal and tracheal swabs. The numbers of wild birds were added to Table 1.

Table 1: replace wild bird species by “bird species” and precise when it is ducks or chickens for each sample collected for backyard poultry.

Corrected

Where is the link for the interactive map? The map in figure 1 is nice but weakly informative because there is no chronological information associated with the different dots and once again the discrimination between ducks and chickens does not appear (for domestic = orange dots)… I hope it is the case in the interactive map.

Originally, the link for the map was placed in the “Data Availability Statement“ paragraph.  It has now been moved to the caption of Fig 1.

What is the correspondence between map 1 and the map in the supplementary data? Not much is missing to make it easier for the reader to understand and follow the spreading of the virus. Maybe it is in the interactive map.

The map in Fig 1 was generated from its interactive version. In the revised manuscript the map in Fig 1 was removed and only the interactive version of the map was provided, as suggested by other reviewers.

In general, the manuscript is well written, but some links are missing between the text and the figures, it has to be improved. Check also the figures in the manuscript (51 wild bird carcasses but 37 described in the table, 70 genomes but 68 in the table.. maybe I’m mistaken).

The number of wild birds (n=51) collected from all locations were supplemented in Table 2, column “bird species”. The Table also includes 71 genomes: 70 H5N8+1 H5N5.

Concerning the phylogenetic trees, in figure 3 the authors discriminate between group A and B.. where are this group in figure 2 tree?

Figures 2 and 3 have been redrawn. In Fig 2 the sub-clades were clearly marked. In addition, groups A and B in Fig 3 have been renamed as 3A and 3B, including the sub-groups, to maintain consistency with Fig 2. The nomenclature has also been corrected in the text.

Line 146: backyard poultry here refers to chickens… is it always the case all along the manuscript?

 Yes, please see our response in the first paragraph.

“Another important aspect of disease transmission to poultry is negligence or loose 228 implementation of biosecurity and preventive measures combined with low levels of sur- 229veillance”. Yes, it is exemplified in the videos provided in supplementary data when you see people among H5N8-infected ducks without any biosecurity equipment (video2) and a guy trying to kick a duck to make it move in video 3 without any biosecurity equipment…. I think you should not present these videos!

All three videos were removed.

Finally, I regret that the authors only analyze in viruses sequence the presence of the polybasic site of HA. What about other known markers? zoonotic potential? 

For instance, in the 2016-2017 "wave" Polish and Czech H5N8 did express a full length PB1-F2 while other european H5N8 mainly expressed a truncated form (non functional, which is exceptional in the avian reservoir). What about these H5N8 viruses? 

All genomes were investigated for presence of molecular markers suggesting increased virulence in mammals, receptor binding preference or antiviral resistence.  See lines 148-149 “No additional mutations conferring antiviral resistance, altering receptor binding preference or increasing virulence or replication ability in mammals were observed“.

I d'ont think that you can obtain any information from the phylogenetic trees presented in supplementary data. Change the format to make them readable.

The phylogenetic trees calculated from European H5N8 sequences collected between Nov 2020-Nov 2021 have been included in the supplementary data.

Reviewer 3 Report

The authors presented a manuscript that seeks to describe the genetic variability of the Czech/2021 (CZE/2021) H5N8 viruses to determine the relationships between strains from wild and domestic poultry and to infer transmission routes between the affected flocks of commercial poultry. The subject is interesting, and the findings are timely with the ongoing HPAI H5Nx epidemic.  

I have some comments, that should be addressed before the manuscript acceptance for publication

Major comments:

1.     In material and methods section, the authors should provide information regarding the time period covered by the study, and the complete procedure to obtain the samples. For instance, to provide information about the number of outbreaks, number of samples per outbreak/farms, among others. Finally, explain if all positive samples were sequenced and considered for phylogenetic analysis, or if a subset was selected. If a subset was selected, explain the process for selection.

2.     Please clarify if phylogenetic analysis were done just for HA gene or the complete virus.

3.     For the phylogenetic analysis, sequences from GISAID database were selected. Please explain the selection process, and how many sequences were used for analysis. 

4.     For the results and conclusions, please consider using the credible intervals to present data for most recent common ancestor information

Minor comments:

 In the introduction, it will be of interest to have some information regarding poultry sector in Czech Republic. Also, will be important to have at least a brief description of influenza virus surveillance activities before and in response to the outbreaks.

Figures 1 and 2.- It is hard to identify the triangles in the figure 1, also (at least for myself) yellow, orange and red dots seems to be very similar in figures 1 and 2. 

Revise carefully the manuscript, since some typos were present

Author Response

Dear editor,

We enclose our response to the reviewers' comments on manuscript ID: viruses-1775847. The vast majority of comments and suggestions have been incorporated into the manuscript.

The main changes in the manuscript suggested by the reviewers are:

  1. Redrawing of Fig 1
  2. Change in Table 1
  3. Removal of the supplementary videos
  4. Inclusion of complete trees of European H5N8 segments in the supplementary data

A detailed response to each comment or suggestion is provided below in italics.

We are grateful for all suggestions as they have greatly improved the quality of our manuscript.

Sincerely,

Alexander Nagy

Rev#3

Major comments:

  1. In material and methods section, the authors should provide information regarding the time period covered by the study, and the complete procedure to obtain the samples. For instance, to provide information about the number of outbreaks, number of samples per outbreak/farms, among others. Finally, explain if all positive samples were sequenced and considered for phylogenetic analysis, or if a subset was selected. If a subset was selected, explain the process for selection.

Most of the information requested were summarised elsewhere in the text:

  1. The time period covered by the study: Lines 45-47 and 100-101.
  2. Number of outbreaks: Lines 102-109.
  3. The number of specimens collected varied considerably from farm to farm and locality to locality. Providing a detailed list of the type and number of samples per bird category in each outbreak would unnecessarily lengthen the manuscript. We did not consider this information relevant to the purpose of the present study.

The specimens for WGS were selected according to Cq values. Line 63.

  1. Please clarify if phylogenetic analysis were done just for HA gene or the complete virus.

The phylogenetic analysis was described in materials and method:. Lines 89-91 “ ML trees (1000 replicates) were calculated separately for each genomic segment and also as a species correlation tree constructed from genomic concatenates“.

  1. For the phylogenetic analysis, sequences from GISAID database were selected. Please explain the selection process, and how many sequences were used for analysis. 

In the phylogenetic analyses we included all available H5N8 sequences collected between Nov 2020 and Nov 2021. The number of sequences included for each dendrogram is summarised in the caption of Fig S3.

  1. For the results and conclusions, please consider using the credible intervals to present data for most recent common ancestor information

We used the LSD method for phylogenetic dating because, according to the authors, its estimation accuracy is similar to that of the most sophisticated methods, while its computing time is much faster. However, it is difficult to estimate the confidence interval for the least square dating method. We used an unpublished strategy implemented in the IQ tree program (http://www.iqtree.org/doc/Dating#dating-an-existing-tree), but, for unknown reason it provided wide intervals. We recalculated the LSD tree several times during the data evaluation including different H5N8 strains and obtained very close dating results. We decided to remove the confidence intervals from the Fig 3.

Minor comments:

 In the introduction, it will be of interest to have some information regarding poultry sector in Czech Republic. Also, will be important to have at least a brief description of influenza virus surveillance activities before and in response to the outbreaks.

The surveillance of AI in the Czech Republic is generally low. Therefore, there were no specific activities other than passive surveillance prior to H5N8 HPAI outbreak identification. Active surveillance in commercial and backyard poultry was only carried out in protective zones as a response of outbreak identification.

This suggestion of the reviewer was not included in the manuscript

Figures 1 and 2.- It is hard to identify the triangles in the figure 1, also (at least for myself) yellow, orange and red dots seems to be very similar in figures 1 and 2. 

The Figures were redrawn. An interactive map was provided for better resolution.

Revise carefully the manuscript, since some typos were present

Done.

Round 2

Reviewer 2 Report

The reviewer would like to kindly congratulate the authors for the work done and the improvement of their manuscript.The article is now acceptable for publication.

Congratulations to the authors.

Reviewer 3 Report

Just minor comments. 

In line 55 change "autopsy" for "necropsy"

Use "italic" for wild birds scientific names